# Retrieval-Based Disease Prediction for Myocardial Injury after Noncardiac Surgery: Leveraging Language Models as Diagnostic Tools

**Namjun Park[1,5], Donggeun Ko[2,5], Dongjun Lee[3], San Kim[1], Jaekwang Kim[1,2,4,5]**

[1]Department of Computer Science and Engineering, Sungkyunkwan University, South Korea,
[2] Department of Applied Artificial Intelligence, Sungkyunkwan University, South Korea,
[3] Department of Electrical and Computer Engineering, Sungkyunkwan University, South Korea,
[4] School of Convergence, Sungkyunkwan University, South Korea,
[5] Convergence Program for Social Innovation, Sungkyunkwan University, South Korea,
{951011jun, seanko, akm5825, saankim, linux}@skku.edu

## Abstract

Predicting Myocardial Injury after Noncardiac Surgery (MINS) is crucial for enhancing patient outcomes, as these injuries significantly affect health and survival rates. This study presents a novel approach for MINS prediction by transforming and converting collected comprehensive pre-operative and intra-operative medical data into a textual description format compatible with Language Models (LM). We employ a Retrieval Based Disease Prediction (RBD) framework, leveraging advanced natural language processing (NLP) techniques to interpret complex patient information. Our results demonstrate that this LM-based approach outperforms traditional machine learning methods. Furthermore, our findings indicate that leveraging LMs with medical data improves predictive performance and potentially enhances patient care and postoperative outcomes. Moreover, the versatility of the RBD framework in adapting to various medical data types highlights its potential as a transformative tool and a stepping stone in healthcare analytics and predictive diagnostics.

## Introduction

Myocardial injury after non-cardiac surgery (MINS) is a critical postoperative complication that profoundly impacts patient health and survival (Smilowitz et al. 2019). The heterogeneous nature of MINS, marked by varied symptoms and diverse patient profiles, presents significant challenges in early detection and prediction (Khan, Alonso-Coello, and Devereaux 2014). Traditional methods for predicting MINS (Szczeklik and Fronczek 2021) often face difficulties due to the various types of data involved, such as binary, numerical, categorical, and textual data making it challenging to scale and interpret them effectively.

In previous studies, Machine Learning (ML) (Nolde et al. 2023; Oh et al. 2023) has been heavily utilized in medical diagnostics, offering novel ways to analyze complex datasets. However, conventional ML approaches often fall short in processing the diverse and unstructured data associated with MINS, primarily due to the necessity for extensive pre-processing and scaling. Furthermore, machine learning methods face difficulty in handling imbalanced data.

In response to these challenges, our study introduces a novel approach, inspired by methodologies previously uti-

**Example of Patient's Textual Description**

The patient, 80-year-old female with height of 155.2cm and weight of 60.1kg, was admitted to general surgery for 'Total Pancreatectomy'. She has a history of diabetes mellitus(DM), hypertension(HTN), cerebrovascular disease(CVD), anemia.
⋮
Mean MAP during hypotensive events was 57.71 mmHg, with an AUT of 768.33 mmHg x minute. 4.0 events with a MAP<50mmHg were observed. Tachycardia(HR>100) was noted for 1.0 minutes. Regular postoperative monitoring and care are advised.

**Definition of Myocardial Injury after Noncardiac Surgery(MINS)**

Myocardial Injury after Noncardiac Surgery(MINS) is defined by at least 1 postoperative cTn(cTn: cardiac troponin) concentration that exceeds the 99th percentile upper reference limit of the cTn assay as a result of a presumed ischemic mechanism in the absence of overt nonischemic causes. Such elevations in cTn must be identified within the first 30 days after surgery but nearly always occur within the first 2 postoperative days.

Figure 1: An illustration showing an example of a patient's medical description in textual format (left) and the medical definition of Myocardial Injury after Noncardiac Surgery (MINS) (right).

lized in the general domain. The Retrieval-Based Frame Prediction (RBF) (Frermann et al. 2023), applied in areas such as news article analysis, serves as a foundation for our innovation. We propose the Retrieval Based Disease Prediction (RBD) framework, an adaptation of the RBF concept, tailored specifically for the medical field. This adaptation represents a significant paradigm shift in medical data processing and analysis.

Our approach transforms multifaceted pre-operative and intra-operative tabular data into coherent text-based descriptions, enabling the use of advanced Language Models (LM) for data interpretation. This method maintains the contextual integrity of medical data and leverages the sophisticated capabilities of LMs in natural language understanding. This paper details the development and application of this innovative method in predicting MINS, illustrating its superiority over traditional models. We explore the transformative impact of converting tabular data into textual descriptions and how this approach, coupled with the RBD framework, opens new frontiers in predictive modeling for postoperative outcomes. Our findings highlight the potential of LMs in medical data analysis and set the stage for more accurate, and efficient approaches in healthcare diagnostics.

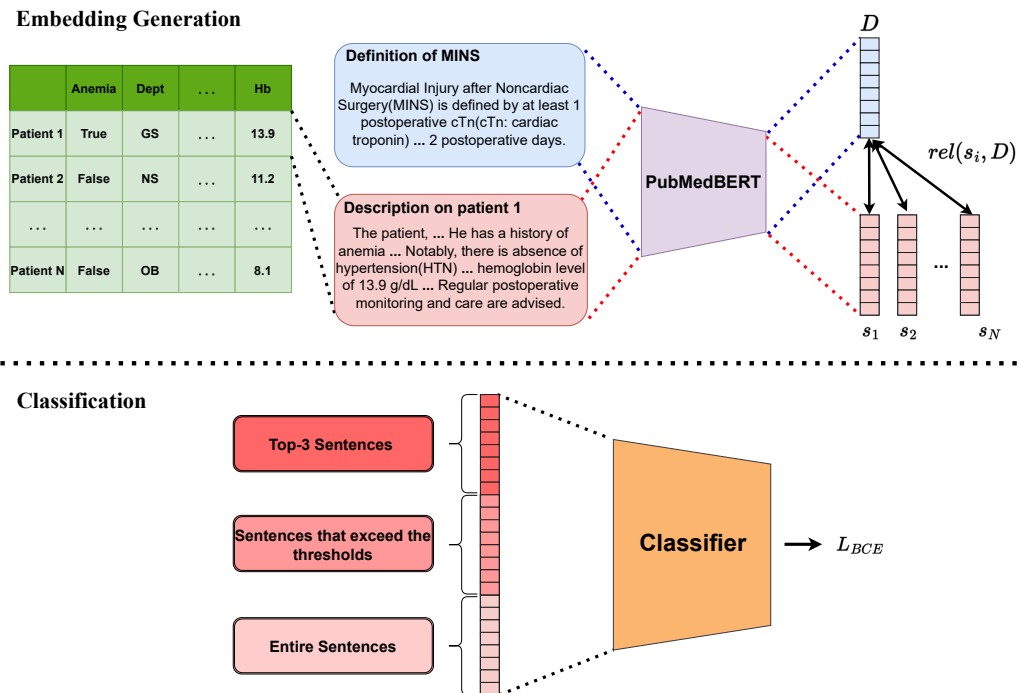

Figure 2: Overview structure of RBD Framework. Our framework is mainly divided into 2 major steps: Embedding Generation and Classification.

## Method

In our methodology, we introduce the RBD framework, an approach inspired by the RBF (Frermann et al. 2023) method previously proposed for predicting news article frames in the general domain. Adapting this concept for disease prediction, we first constructed a comprehensive dataset with medical experts' assistance. This dataset transformed tabular patient data into rich textual descriptions, enhancing the interpretability of complex medical information. Furthermore, we meticulously crafted the definition of MINS (Ruetzler et al. 2021; Nolde et al. 2023), the condition our framework aims to predict, with medical experts to ensure precise and clinically relevant representation. These efforts have yielded a robust foundation for our RBD framework to interpret and analyze clinical data for predictive purposes effectively. Figure 1 shows an example of the textual description of a patient and the definition of MINS.

$$
\begin{aligned}
h_i^s &= emb(s_i) \\
h_D &= emb(D) \\
rel(s_i, D) &= cos(h_i^s, h_D)
\end{aligned}
\tag{1}
$$

We embed the definition of MINS, denoted as $D$, and sentences from the input patient description $s_1, \ldots, s_n$ in a joint space to retrieve sentences that are most relevant to the disease definition embedding. Let $h_i^s$ represent the embeddings of sentence $s_i$, the relevance $rel$ of $s_i$ to $D$ is quantified by their cosine similarity. Embeddings are derived using PubMedBERT (Gu et al. 2021). Unlike models pre-trained on general domain data and then continually trained on specialized medical data, PubMedBERT chose a different approach due to the limited performance of aforementioned approach. It is instead trained from scratch exclusively on medical texts, which has been shown to yield better performance when compared to other medical domain pre-trained models.

A linear classifier is then trained to predict the presence or absence of a disease based on the most relevant sentences as determined by our measure. We introduce five input channels: channels $(1) - (3)$ consist of the three sentences with the highest RBD relevance for the disease; channel $(4)$ includes all sentences with a relevance threshold $\theta > 0.3$; and channel $(5)$ contains the entire patient description. Each channel is encoded using PubMedBERT, and the embeddings from the final hidden state are concatenated and fed into the classifier. Figure 2 provides a visual representation of the composition of the input channels within the RBD framework, showing the specific structure and integration of each channel into the overall predictive model. The importance of different channels is also evaluated by ablating the impact of the threshold sentence channel $(4)$, referred to as RBD-t.

## Experiments

### Dataset

We collected a dataset for the prediction of MINS, which comprises anonymized medical information of 502 actual patients. Among these, 40 cases, which is approximately 7.9% of the total patients, experienced myocardial injury af-

ter non-cardiac surgery. This highlighted a significantly imbalanced incidence rate, posing a major challenge for enhancing the performance of our model.The presence of such a pronounced class imbalance is particularly problematic in medical predictive modeling because it can lead to a bias towards the majority class, reducing the model's sensitivity to detect less frequent, yet clinically significant, events like myocardial injuries post-surgery. This imbalance can skew the model's performance metrics, giving a false sense of accuracy while failing to identify critical outcomes accurately. Therefore, addressing this imbalance is crucial not only for achieving a more accurate and reliable model but also for ensuring that the model can effectively contribute to improving patient care by accurately identifying those at risk of MINS. Despite the small amount of data and the highly imbalanced class distribution, our goal was to develop a model that could still perform effectively, recognizing the importance of accurately predicting such critical medical events despite their rarity in the dataset.

The dataset includes information that can be obtained pre-operatively, such as the patient's age, height, weight, surgical department, surgery name, and any underlying conditions. It also contains intra-operative details like the duration of surgery, the total number of hypotensive events during surgery, the average duration of each hypotensive event, among others. Utilizing this information, we collaborated with a medical expert to construct textual descriptions for each patient that are akin to those found in actual patient charts in the medical field, ensuring a comprehensive and well-represented depiction of their medical profile.

### Implementation Details

For machine learning methods as baselines, we utilize the collected tabular data and did not alter its structural format. The categorical data are processed using label encoding, converting categorical labels into numerical format. Then, these encoded numerical data are standardized by applying z-score normalization. For methods utilizing language models, we leverage embeddings derived from textual descriptions. These embeddings encapsulate rich contexts within the descriptions, transforming them into a dense, machine-readable vector format. These vectors serve as inputs to our models, enabling them to discern patterns and associations pertinent to MINS prediction.

RBD framework enhances predictive analytics by using a relevance-based input channel, aligning patient descriptions with specific disease definitions, like MINS. This approach allows RBD to focus on the most pertinent features for accurate prediction, improving its overall performance.

### Comparison Models

We compared our method against 1. **RandomForest** (Breiman 2001) baseline; 2. **XGBoost** (Chen and Guestrin 2016) baseline; 3. **BERT-base** (Devlin et al. 2018); 4. **SciBERT** (Beltagy, Lo, and Cohan 2019) fine-tuned for binary classification; 5. **ClinicalBERT** (Huang, Altosaar, and Ranganath 2019) fine-tuned for binary classification; 6. **BioBERT** (Lee et al. 2020) fine-tuned for binary classification

| Model | Precision | Recall | F1 |
|---|---|---|---|
| RandomForest | **0.63**($\pm$0.350) | 0.17($\pm$0.100) | 0.27 |
| XGBoost | 0.48($\pm$0.290) | 0.25($\pm$0.112) | 0.33 |
| BERT-base | 0.27($\pm$0.086) | 0.42($\pm$0.170) | 0.33 |
| SciBERT | 0.26($\pm$0.093) | 0.50($\pm$0.194) | 0.34 |
| ClinicalBERT | 0.27($\pm$0.073) | 0.50($\pm$0.137) | 0.35 |
| BioBERT | 0.30($\pm$0.063) | 0.49($\pm$0.172) | 0.37 |
| **RBD** | 0.60($\pm$0.168) | **0.57**($\pm$0.170) | **0.59** |
| RBD-t | 0.41($\pm$0.144) | 0.44($\pm$0.203) | 0.42 |

Table 1: MINS prediction results of machine learning baseline models (top) language models, RBD (middle), and an ablation of RBD channel (bottom). We report averaged precision and recall across the 10 different seeds with standard deviation (in brackets), and their harmonic mean (F1). Best and runner-up performance is in bold and underlined, respectively.

### Results

Table 1 demonstrates MINS prediction results of machine learning and language models baselines as well as our RBD framework. Our study evaluated the performance of various machine learning and language models in predicting MINS. To establish the robustness of our model, we conducted experiments across ten different seeds.

Language models consistently outperformed traditional ML approaches, indicating their superior capability in managing the complexity of medical data with respect to F1 scores. Interestingly, while ML models demonstrated relatively higher precision, this did not translate into overall better performance, heavily due to the imbalanced nature of the dataset which appears to bias the precision metric. Additionally, while ML models demonstrated relatively higher precision compared to language models, they also exhibited a higher standard deviation. This significant variance in performance depending on the seed value indicates that ML models are less robust, as it points to considerable inconsistency in their precision. This inconsistency in precision, especially given the high variability influenced by seed values, raises concerns about the reliability of ML models in accurately identifying actual cases of MINS, which is a fundamental limitation in their clinical applicability.

Furthermore, we found that BERT-based models pretrained on medical data outperformed the BERT-base model. This can likely be attributed to the medical domain-specific vocabulary present in the patient descriptions, which suggests that a domain-specific representation is crucial for enhancing model performance.

Most notably, our proposed RBD framework showed superior performance when compared to other models. Additionally, we conducted an ablation study by removing sentences from the input channel that exceeded a threshold value. The resulting RBD-t model exhibited a reduction in performance compared to the original RBD model. Even with this reduction in performance, the RBD-t model still exhibited superior performance compared to ML models and other language models. This outcome suggests that the pres-

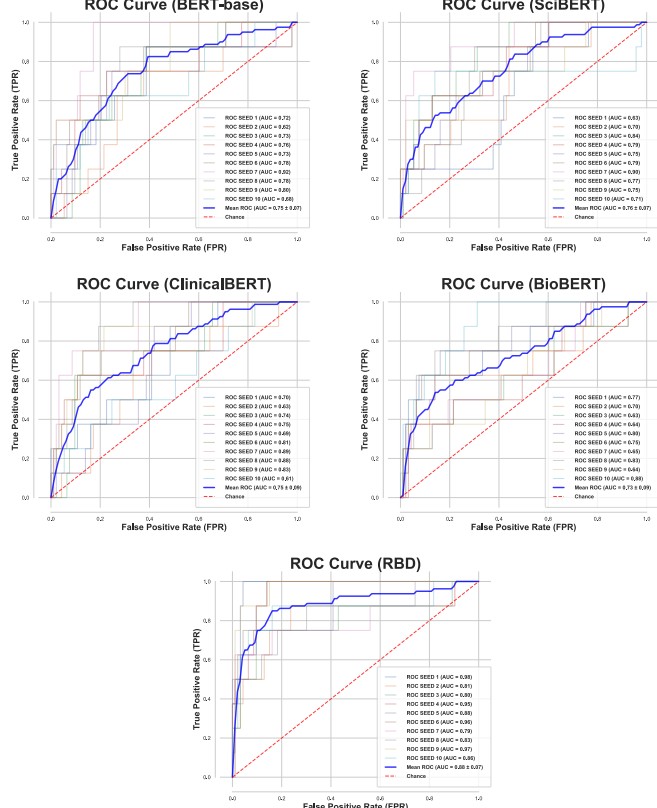

| Top-K | Precision | Recall | F1 |
|---|---|---|---|
| 1 | 0.40(± 0.174) | 0.46(± 0.177) | 0.43 |
| 2 | 0.42(± 0.164) | 0.49(± 0.189) | 0.45 |
| **3** | **0.60**(± 0.168) | **0.57**(± 0.170) | **0.59** |
| 4 | 0.50(± 0.152) | 0.51(± 0.172) | 0.50 |
| 5 | 0.53(± 0.144) | 0.55(± 0.170) | 0.54 |

Table 2: Hyperparameter analysis of RBD using Top-K sentence selection. The table demonstrates precision and recall across 10 different seeds with standard deviation (in brackets), and their harmonic mean (F1) for varying K values of Top-K. Best and runner-up performance is in bold and underlined, respectively.

perimental results are shown in Table 2. It was observed that the best performance was achieved when the RBD model, as presented above, adopted a K value of 3. Even with different K values, it was confirmed that in all cases, our model outperformed other machine learning models' or language models' performance. This underscores the effectiveness of our method of constructing input channels, as opposed to simply using tabular data or descriptions, in enhancing performance.

## Summary

**Conclusion** We proposed the Retrieval Based Disease (RBD) Prediction framework for disease prediction, demonstrating through experimentation that this method outperforms conventional machine learning techniques and other language model-utilizing approaches. The key contribution of our study is to increase the predictive performance of medical data with heavy class imbalances in disease prediction, which are typically challenging for conventional ML techniques. We employed language models that utilize detailed descriptions rich in patient information to bolster predictive performance. Our findings confirm that employing language models significantly improves performance over traditional ML methods in scaling tabular data inclusive of various data types. Moreover, we have established that the RBD approach advances beyond the mere application of language models, offering superior diagnostic performance. This underscores the potential of RBD in leveraging language models to effectively interpret complex medical datasets for improved diagnostic tools.

**Future Works** While current research has focused exclusively on utilizing RBD framework for datasets about MINS prediction, future endeavors aim to extend this approach to open datasets for broader disease prediction.

Additionally, we plan to expand beyond binary classification for a single disease prediction. Our objective includes developing datasets capable of handling multi-label classifications, allowing for the prediction of multiple diseases a patient may have. This holistic approach will enhance the versatility and applicability of our predictive models in diverse medical scenarios.

Figure 3: ROC-AUC visualization for different language models, including BERT-base, SciBERT, BioBERT, ClinicalBERT and RBD across 10 different seeds. The red dashed line on the graph indicates the performance of random guessing (AUC=0.5).

ence of the input channel contributes significantly to the enhancement of the model's performance.

We also performed a performance comparison between other language models and our proposed RBF through ROC-AUC visualization, as shown in Figure 3. To evaluate the robustness of the model, experiments were carried out with 10 different seed values, and for a fair selection of seeds, continuous values from 1 to 10 were used to set the seeds. The mean AUC values from 10 different seed values for each language model were as follows: BERT-base model showed $0.75(\pm0.07)$, SciBERT model showed $0.76(\pm0.07)$, ClinicalBERT model showed $0.75(\pm0.08)$, and BioBERT model showed $0.73(\pm0.09)$. In contrast, our proposed Framework, RBD, recorded $0.88(\pm0.07)$, demonstrating superior performance over the existing models. Through this, it was revealed that our proposed RBD Framework exhibits higher diagnostic accuracy than methods utilizing only pre-trained language models in the medical domain, and its potential for use as a medical diagnostic tool was confirmed.

In addition, we conducted experiments by setting various K values as a hyperparameter for selecting the sentences with the highest relevance to the disease definition. The ex-

## Acknowledgement

This research was supported by the MSIT(Ministry of Science and ICT), Korea, under the ICAN(ICT Challenge and Advanced Network of HRD) support program(IITP-2024-RS-2023-00259497) supervised by the IITP(Institute for Information Communications Technology Planning Evaluation) and This work was supported by Institute of Information communications Technology Planning Evaluation (IITP) grant funded by the Korea government(MSIT) (No.RS-2023-00254129, Graduate School of Metaverse Convergence (Sungkyunkwan University)).

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
