# OpenReview forum: "Retrieval-Based Disease Prediction for  Myocardial Injury after Noncardiac Surgery: Leveraging Language Models as Diagnostic Tools"
_AAAI.org/2024/Spring_Symposium_Series/Clinical_FMs — AAAI 2024 SSS on Clinical FMs_

### Official Review · Reviewer_RopL · 2024-02-13

**Rating:** 4
**Confidence:** 5

**Review:**

This paper explains an approach to convert tabular data into textual descriptions that can be parsed by a language model for comparison against a specific medical definition.

Strengths:
1. The paper presents an interesting idea to supplement LM models with clinical definitions to inform model predictions.
2. Multiple experiments were performed. Results were averaged and standard deviations were included.

Weaknesses:
1. Paper contributions are misleading. Paper says, "approach transforms multifaceted pre-operative and intra-operative tabular data into coherent text-based descriptions, enabling the use of advanced Language Models (LM) for data interpretation." This seems to imply that a model is used to output text descriptions from static data but really these descriptions are developed using medical experts. Also the conclusions say, "The key contribution of our study is to increase the predictive performance of medical data with heavy class im- balances in disease prediction". Class imbalance is not discussed as an objective in the introduction and it unclear how the method was designed for this purpose in mind.
2. More explanation of the RBF framework is needed. What is it? How does your method differ/expand on this approach?
3. Paper contributions as explained do not seem to present a novel methodology. The text and definition are developed by medical experts and then just parsed through an existing LM where the output embeddings are compared using cosine similarity. It seems that the main contribution is the approach to group sentence inputs to a downstream classifier for prediction rather than directly using an LM model fine-tuned for binary classification. Perhaps the embedding approach for grouped sentences is novel? More explanation and a perhaps refocused message is needed.
4. Based on confidence intervals the results do significantly improve over baselines in terms of precision and recall.
5. Explanation of baseline LM training is vague.
6. How was the relevance threshold selected? This should be a hyperparameter that is tuned based on a validation set and not manually selected based on model performance. More explanation is needed here.

Questions:
1. Is the definition of MINS provided to the other baseline LMs? If not, it would be difficult to compare how this approach compares to other baseline LMs and assess whether the downstream classifier is necessary.
2. What is the purpose of the different inputs channels? Is it to create embeddings over a set of grouped sentences rather than individual sentences? Did you experiment with just inputing the embedding over the entire patient description? This would be similar to using an LM binary classifier baseline.

Other:
1. I would recommend emphasizing the value of this approach to adapt to new/changing definitions easy without needing to relabel the data and retrain a model. That is a very valuable approach in healthcare.

---

### Official Review · Reviewer_PdWD · 2024-02-22
**Strong manuscript**

**Rating:** 9
**Confidence:** 4

**Review:**

In reviewing this manuscript, it is clear the authors can intellectually articulate the focus of their study. They were able to explain the rationale of their research as well as their results.

I would encourage the authors to discuss the results of their RBD model in the abstract.” At present, the abstract states their "LM-based approach outperforms traditional machine learning methods" but does not state any results.

Overall, I believe this is a strong work from the authors.

---

### Official Review · Reviewer_YG1g · 2024-02-23
**Retrieval-Based Disease Prediction for Myocardial Injury after Noncardiac Surgery: Leveraging Language Models as Diagnostic Tools**

**Rating:** 8
**Confidence:** 3

**Review:**

- Myocardial injury after non-cardiac surgery (MINS) is a postoperative complication presented in diverse profile patients and marked by varied symptoms. Thus, its early prediction is a challenge and ML approaches, even heavily utilized, present some difficulties in processing unstructured and imbalanced data.
- Authors propose a framework based on the one presented in https://arxiv.org/pdf/2306.02052.pdf (RBF), called Retrieval Based Disease (RBD) Prediction framework. Their approach transforms multifaceted pre-operative and intra-operative tabular data into coherent text-based descriptions, enabling the use of Language Models (LM) for data interpretation.
- The authors face an imbalanced dataset of MINS, with only 7.9% of the total patients experiencing MINS. They compare their framework with a RandomForest, a XGBoost, a BERT-base and a SciBERT, ClinicalBERT and BioBERT fine-tuned for binary classification. They show that their RBD framework outperformed the ML approaches and the LM ones. Even with an ablation study, RBD-t showed better performance than the others (but the original RBD).

The presented framework is a variation of a pre-existing one (RBF) and combines it with other Languange Model for Biomedical domain (PubMedBERT). However, the methods seem to be properly adapted and a good choice to the problem to be tackled, with the corresponding citation. The framework is correctly presented in Figure 2 and the results of the few experiments presented clearly in Tables 1 and 2.

The presented results show a good improvement compared with the other baselines, which seem to be diverse and relevant enough; first, two widely used ML techniques and then 4 domain specific language models. Even the ablated version of the framework (RBD-t) overperforms the other baselines, despite performing worse than the not ablated RBD.

Even though it uses pre-existing methods, the need of adaption of it to imbalanced datasets is essential and the significance of having early detection techniques without the need of post-operative data is valuable.

Regarding to format and clarity, the work is well presented and clearly structured, having not found major errors in the writing. The trivial next steps are presented as future work, with the aim of making the framework more general by extending it to multi-label datasets.

COMMENT:

- If I am not wrong, the only innovation here is the adaptation of RBF to medical data by using a specific BERT model for biomedical NLP tasks, named PubMedBERT. Right?
- What is the explanation of K=3 showing the best results?

---

### Official Review · Reviewer_9CR7 · 2024-02-23
**Retrieval-Based Disease Prediction for Myocardial Injury after Noncardiac Surgery: Leveraging Language Models as Diagnostic Tools**

**Rating:** 8
**Confidence:** 3

**Review:**

Brief Overview of the Paper
The paper presents a novel approach to predicting Myocardial Injury after Non-cardiac Surgery (MINS) using a Retrieval Based Disease (RBD) prediction framework. This method innovatively addresses the challenge of analyzing complex, unstructured clinical data by transforming it into a coherent text description for improved decision-making accuracy. The authors claim their Language Model (LM)-based model outperforms traditional Machine Learning (ML) models, offering promising results that suggest further validation is needed in larger datasets or new clinical scenarios with different disease prevalence.

Quality
Technical Soundness and Description
  The methods and accompanying figures presented by the authors are technically sound and well-described, providing a clear overview of the proposed RBD framework and its advantages over traditional ML approaches. The incorporation of various types of clinical data into a unified text-description model showcases a significant advancement in handling and interpreting complex data for disease prediction.

Addressing Data Imbalance:
  While the authors acknowledge the challenge of working with an imbalanced dataset due to low disease prevalence, the paper would benefit from a more detailed discussion on how to address this issue. Suggested improvements include implementing resampling strategies, such as oversampling the under-represented disease class, to mitigate the imbalance's impact. Additionally, incorporating Receiver Operating Characteristic (ROC)-Area Under Curve (AUC) comparisons among the models could provide deeper insights into their performance, especially in managing false positives and negatives in an imbalanced dataset context.

Clarity and Justification
 The statistical methods used in the paper are clearly explained and justified. The authors' approach to analyzing the data and the results presented are sound, offering a solid foundation for their conclusions.

Recommendation & Significance
This paper introduces a compelling and innovative approach to disease prediction using LM-based models, addressing significant challenges in analyzing unstructured clinical data. While the results are promising, addressing the highlighted areas for improvement, particularly around data imbalance and statistical analysis depth, could significantly enhance the paper's impact and validity. Further validation of the proposed model in broader clinical settings is recommended to substantiate its effectiveness and applicability in real-world scenarios.

Additional Comments
Strengths:
- The novel RBD framework represents a significant innovation in leveraging LM-based models for disease prediction, especially in dealing with complex, unstructured clinical data.
- The initial findings suggesting superior performance of the RBD framework over traditional ML models are promising and warrant further investigation.
Weaknesses:
- The paper lacks a comprehensive strategy to address the issue of data imbalance, which is crucial for validating the model's effectiveness across different clinical scenarios.
- The statistical analysis would benefit from more detailed comparisons between models and a clearer justification for methodological choices, such as the selection of k-values.

Recommendations for Improvement:
To strengthen the paper's statistical analysis, it would be beneficial to include a direct statistical comparison of the models' recall and F-1 scores. This comparison could highlight the proposed model's effectiveness more clearly against traditional ML approaches. Additionally, the rationale behind selecting a k-value of 5 for certain analyses requires further explanation. Expanding on this decision could enhance the reader's understanding of the methodology and its implications for the study's findings.